# Evidencing the Influence of the COVID-19 Pandemic and Imposed Lockdown Measures on Fitness Status in Adolescents: A Preliminary Report

**DOI:** 10.3390/healthcare9060681

**Published:** 2021-06-05

**Authors:** Mirela Sunda, Barbara Gilic, Ivan Peric, Anamarija Jurcev Savicevic, Damir Sekulic

**Affiliations:** 1Antun Gustav Matos High School, 31400 Dakovo, Croatia; mirela.sunda@skole.hr; 2Faculty of Kinesiology, University of Zagreb, 10000 Zagreb, Croatia; barbaragilic@gmail.com; 3Faculty of Kinesiology, University of Split, 21000 Split, Croatia; 4Faculty of Dental Medicine and Health, Josip Juraj Strossmayer University of Osijek, 31000 Osijek, Croatia; ivan.peric@fdmz.hr; 5Teaching Institute of Public Health of Split Dalmatian County, 21000 Split, Croatia; anamarijajs@gmail.com; 6Department of Health Studies, University of Split, 21000 Split, Croatia

**Keywords:** education, exercise, health fitness, pandemic, physical capacities, youth

## Abstract

It was established that the COVID-19 pandemic resulted in decreased physical activity levels, potentially leading to reduced physical fitness. The aim of this research was to investigate the impact of the COVID-19 lockdown on fitness indices in adolescents. We observed high school adolescents (33% girls; aged 15–17 years) divided into two cohorts. The first cohort (control; *n* = 48) included adolescents who were in school during the 2018/2019 year, while the second cohort (lockdown; *n* = 66) was observed during the school year 2019/2020 when COVID-19 lockdown measures were imposed. Variables included body height, body mass, BMI, sit-ups, and the 600 m dash. Both cohorts were tested at the beginning and at the end of the observed school years. A factorial analysis of variance for repeated measurements evidenced an improvement in sit-ups and the 600 m dash in the control, and a decrease of the same capacities in the lockdown cohort. In the lockdown cohort, a decrease in muscular fitness was more evident in boys than in girls. No differential changes between cohorts were evidenced in anthropometrics. Results suggest that the COVID-19 lockdown negatively influenced muscular fitness status in adolescents, especially in boys.

## 1. Introduction

The COVID-19 pandemic, declared at the beginning of 2020 due to the fast spread of the SARS-CoV-2 virus, has dramatically changed the lives of people worldwide [1]. The major strategy for reducing the incidence of new cases of the disease was the implementation of social distancing measures and lockdown (home confinement), which included closures of all places where a large number of people can gather (schools, universities, churches, training facilities, and sports clubs) [2]. The imposed restrictions resulted in reduced movement opportunities and consequently decreased physical activity levels (PAL). Namely, numerous studies reported a significant decline in PAL as a result of the COVID-19 lockdown [3]. The decrease in PAL as a result of the lockdown was particularly alarming in adolescents [4].

Specifically, a study conducted on adolescents from Ireland reported that 50% of adolescents included in the study exhibited decreased PAL during the COVID-19 lockdown [5]. Similarly, Spanish adolescents presented decreases in both moderate and vigorous PAL by 29.5% and 18.3%, respectively, during the lockdown [6]. Reduced PAL and increased sedentary behavior were confirmed in adolescents from Australia, Canada, Spain, Italy, and Germany [4,7,8,9,10,11,12]. Results from studies conducted on adolescents from Croatia, and Bosnia and Herzegovina evidenced decreased PAL as a consequence of restrictions related to the COVID-19 pandemic [13,14,15,16]. Specifically, the PAL of Croatian adolescents decreased from 2.97 ± 0.61 to 2.63 ± 0.68 (as measured by the Physical Activity Questionnaire for Adolescents, or, PAQ-A) [14]. The PAQ-A is a questionnaire that assesses PAL during the last seven days, with results ranging from one to five, representing low and high PAL, respectively. Similarly, 50% of adolescents from Bosnia and Herzegovina experienced sufficient PAL before the COVID-19 lockdown, while only 24% achieved sufficient PAL during the pandemic [15]. Consequently, there is a global concern that such a dramatic decrease in PAL will have detrimental consequences on the physical fitness of youth.

Physical fitness represents a state of several bodily functions, including musculoskeletal, cardiorespiratory, and metabolic functions, that are crucial for performing physical activities and physical exercise [17]. It refers to numerous physical capacities, including muscular strength, speed, agility, flexibility, coordination, and muscular fitness [18]. While a higher level of physical fitness is positively correlated with health status, achieving and maintaining an adequate level of physical fitness is especially important in adolescence, as it is a life period in which the individual undergoes growth and development processes, the development of most motoric capacities occurs, and adulthood health is determined [19,20,21]. Importantly, physical activity has a direct impact on physical fitness. For example, a study by Júdice et al. [22] reported that moderate to vigorous physical activity was positively associated with physical fitness in girls and boys, and they suggested that engaging in more physical activity and reducing time spent in sedentary behavior will produce benefits on health-related fitness in adolescents [22]. Accordingly, a Norwegian school-based intervention on physical activity resulted in increased PAL and improved physical fitness in adolescents, with an emphasis on cardiorespiratory fitness [23]. Similarly, after 16 weeks of physical activity promotion, Korean adolescents increased their physical fitness levels [24].

It is clear that COVID-19 induced a significant decrease in adolescents’ PAL, which is recognized as a global problem [3]. Primarily, authors regularly noted that such a dramatic decline in PAL could have detrimental effects on various health indices, including physical fitness [25,26]. However, as far as we are aware, no study has yet directly investigated changes in physical fitness as a consequence of restrictions related to the COVID-19 pandemic. Therefore, the aim of this study was to investigate the impact of the COVID-19 lockdown on several components of physical fitness in adolescents from Croatia. We hypothesized that measures of social distancing and lockdown imposed as a result of the COVID-19 pandemic would negatively influence physical fitness variables (anthropometric and motor-endurance status) in adolescents irrespective of gender.

## 2. Materials and Methods

### 2.1. Participants and Study Design

Participants included in this study were 114 adolescents (76 girls, 38 boys), who were high school students in one high school in Đakovo, Croatia. Participants were 16.25 ± 0.55 years of age at the study baseline. All participants attended regular physical education (PE) classes two times a week in the same facility and were instructed by the same PE teacher. All students were healthy during the study period, and they did not have any injury or illness that would have prevented them from participating in the physical education classes. This study was part of another longitudinal investigation which was initiated in 2019, and all participants were informed about the aims and protocols of the study, and their parents signed informed consent forms. In brief, the original study titled “Physical activity, substance misuse, and factors of influence in adolescence” aimed to establish changes in health-related variables (i.e., PAL, misuse of psychoactive substances, fitness status), which allowed us to observe adolescents even during the COVID-19 pandemic. This research was approved by the Ethical Board of the University of Split, Faculty of Kinesiology (approval number: 2181-205-02-05-20-004).

This prospective study consisted of two testing waves conducted on two cohorts (generations): (i) the generation before the COVID-19 pandemic (the 2018–2019 school year; *n* = 48, 15.35 ± 0.31 years at baseline), hereafter referred to as the control cohort/group, and (ii) adolescents that were affected by the COVID-19 pandemic (the 2019–2020 school year; *n* = 66, 15.65 ± 0.51 years at baseline), referred to as the lockdown/COVID cohort/group (Figure 1).

Both cohorts were tested on study variables (see subsequent sections for details) at the beginning of the school year (pre-testing; late September-early October of 2018 or 2019, depending on the generation) and again at follow-up (post-testing; late May-early June of the subsequent year, either 2019 or 2020). All testing procedures were part of regular testing on anthropometric and physical performance variables included in the school PE program. The control group participated in regular PE classes twice a week throughout the school year. Lockdown generation students attended regular PE classes between September 2019 and February 2020, while from March until the end of school year in late May 2020 (the period of imposed measures of social distancing and lockdown), students followed instructions about PE topics over online learning platforms and had to individually accomplish certain physical exercising tasks. The strictest lockdown in Croatia was implemented from the 19th of March and lasted until the 27th of April 2020. Schools and universities were closed, public gatherings were prohibited, and cafes, restaurants, fitness centers, shopping centers, and churches were closed. Playground and parks were also closed; however, there was no strict prohibition of outdoor individual training (i.e., running, cycling). However, post-testing for the lockdown group was performed in regular circumstances (in the presence of a PE teacher).

### 2.2. Variables

In Croatian high schools, physical fitness is typically evaluated using a wide range of physical fitness tests. However, due to the specific situation, the lockdown cohort was tested by two fitness tests (at post-testing), so in this study we observed: (i) sit-ups for 30 s (SU30) and (ii) running 600 m (600 m). SU30 was used to evaluate muscle endurance of the abdominal region, and it was conducted in the school gym. Participants had their palms on their thighs, their knees were bent, and their feet were fixed during the test exercise (with their partner sitting on their feet and holding the examiner’s legs with both hands). Participants had to lift their torso and slide their palms towards their knees. The result of the test was presented as the number of valid repetitions of sit-ups in 30 s. The 600 m test was used to evaluate aerobic/anaerobic capacity. Participants had to run a distance of 600 m in a concrete athletic field, on a 300 m oval track. The results were recorded in seconds required to run the given distance. Anthropometric variables included body height (BH), body mass, and calculated body mass index (BMI) (BMI = weight (kg)/height^2^ (m)). All tests previously displayed appropriate reliability and validity in a similar sample of participants [27,28]. All tests were administered by an experienced evaluator, i.e., the PE teacher who is the first author of this paper.

### 2.3. Data Analysis

Normality of the distributions was checked by the Kolmogorov Smirnov test and descriptive statistics included means and standard deviations. In order to identify the effects of lockdown measures on studied variables we used a multifactorial analysis of variance (ANOVA) for repeated measurements, with consecutive Scheffe post-hoc tests. Specifically, two blocks of ANOVA calculations were done. First, we included the total sample of participants (i.e., both cohorts) and calculated repeated measurement ANOVA with “cohort” as the grouping variable (e.g., measurement × cohort). In the next phase, an additional block of ANOVA calculations was done for the lockdown cohort exclusively, with “gender” as the grouping variable (e.g., measurement × gender). To evaluate the effect sizes (ES), partial eta squared values (η^2^) were reported with values >0.02; >0.13; >0.26 representing the low, medium, and high ES, respectively [29]. Mean-changes, and 95% confidence intervals (95%CI) were reported when appropriate.

All calculations were done by Statistica ver. 13.5 (Tibco Inc., Palo Alto, CA, USA) and a statistical significance of *p* < 0.05 was applied.

## 3. Results

ANOVA for repeated measurements (baseline vs. follow-up) calculated for the total sample with the grouping variable cohort (control vs. lockdown cohort) showed a significant “measurement” effect for body mass (F-test = 4.37, *p* < 0.05; small ES) and run-600 m (F-test = 40.68, *p* < 0.001; large ES). The interaction effect (cohort x measurement) was significant for sit-ups (F-test = 67.11, *p* < 0.001; large ES) and run-600 m (F-test = 61.97, *p* < 0.001; large ES) (Table 1).

Descriptive statistics evidenced that the lockdown group achieved lower results in fitness tests (e.g., sit-ups and run-600 m) compared to the control group at follow-up testing. A post-hoc analysis indicated differential changes in sit-ups and run-600 m between cohorts. In brief, both variables positively changed in the control group from baseline to follow-up testing (mean-change, 95%CI: sit-ups: 7.79, 6.05–9.53; run-600m: 4.40, 4.26–13.06). On the other hand, the negative changes were evidenced for the lockdown group in sit-ups (mean-change, 95%CI: 7.59, 5.51–9.67) and run-600 m (mean-change, 95%CI: 41.97, 33.99–49.95) (Table 2).

When examined specifically for the lockdown group, ANOVA for repeated measurements (with gender as the group), evidenced a significant effect for “measurement” in body mass (F-test = 4.23, *p* < 0.05; small ES), sit-ups (F-test = 45.44, *p* < 0.05; large ES), and run-600 m (F-test = 105.11, *p* < 0.001; large ES). A significant interaction effect (measurement x group) was found for sit-ups (F-test = 11.08, *p* < 0.01; medium ES) (Table 3).

Post-hoc analyses indicated a significant decrease from baseline to follow-up testing of sit-ups in females (mean change, 95%CI: 5.59, 3.27–7.91), with a more evident decrease in males (mean change, 95%CI: 12.0, 8.13–15.87) in the lockdown generation (Table 4).

## 4. Discussion

There are several main findings in this study. First, during the observed period muscular fitness increased in the control, but declined in the lockdown cohort, with a more evident decline in boys. Meanwhile, changes in morphological variables were similar in the control and lockdown cohorts. Therefore, our initial study hypothesis may be partially accepted.

Although preliminary, our results demonstrate potentially negative effects of the COVID-19 pandemic on changes in muscular fitness in adolescents. Namely, when compared to the previous generation of students who were not affected by social distancing and lockdown, the COVID-19 generation experienced negative trends in observed variables of muscular and cardiorespiratory fitness. Indeed, different opinion papers have warned about the expected negative consequences of reduced physical activity caused by the COVID-19 pandemic on physical fitness [25]; however, to the best of our knowledge, our study is the first to directly confirm such negative projections. There is no doubt that the most important influencing factor on such negative changes should be found in the decrease in physical activity in adolescents during the period of the COVID-19 pandemic, which has been repeatedly reported [15,30]. Specifically, a lack of physical activity has a direct impact on decreased muscular fitness, especially among adolescents who are undergoing the process of physical growth and development [19].

Irrespective of the fact that negative changes in the COVID-19 generation were evidenced for the total sample, our analyses indicated a more negative trend in muscular fitness among boys. To explain this finding, a physiological and biological background of the studied sample (e.g., adolescents) should be briefly presented. First, it is well known that physical activity, training, and exercise have an impact on hormonal response, especially with respect to anabolic responses [31]. Most importantly, physical exercising may stimulate the release of insulin-like growth factor-1, testosterone, and growth hormone, resulting in better adjustments to the exercise effort of various body functions, including the musculoskeletal system [32]. However, boys (men) exhibit more efficient anabolic pathways (with emphasis on testosterone) associated with physical exertion than girls [33]. Since lack of physical activity and exercise leads to lowering the level of anabolic hormones, the more evident and rapid decline in fitness capacities in boys evidenced herein is understandable [32].

The second factor that may contribute to differences in the decline of muscular fitness between sexes is that boys decreased their PAL during the pandemic to a greater extent than girls did [4,15]. Although the changes of PAL were not evidenced in this study, previous studies in Croatia and neighboring countries consistently highlighted a more evident decrease of PAL for boys than for girls [4,13,14,15]. Therefore, it is logical to conclude that PAL in boys during the imposed measures of lockdown was alarmingly lower than in boys of the same age a year before, when children lived in regular circumstances. Taken together, the lack of physical activity and the reduced metabolic and hormonal levels could underlie the finding that boys decreased their muscle fitness to a greater extent than girls.

Finally, as the authors are actively involved in PE classes (as PE teachers), one additional reason for the greater decline in muscular fitness among boys needs attention. During the pandemic period, PE classes were organized online using online platforms and mobile applications. In general, teachers provided different instructions and various forms of exercise to their students who had to perform them individually [34]. From our perspective, girls were more prone to follow recommended exercise programs and, in most cases, practiced according to the teacher’s suggestions. In contrast, this was not the case for boys. Thus, it is possible that during lockdown, muscle fitness levels in girls decreased less dramatically due to their higher involvement in exercise programs specifically designed in exchange for regular PE classes.

We found no evidence that the COVID-19 pandemic and imposed measures of social distancing and lockdown significantly influenced the anthropometric/morphological status of adolescents. Although we observed only a few variables, this finding could be explained considering the life period of our participants. Specifically, this study included adolescents aged 15 to 17 years who were mostly in the period of growth and physical maturation (puberty). This life period is very dynamic and characterized by rapid changes in body size, shape, and composition [35]. Therefore, it is unlikely that under conditions of normal growth and development (i.e., all participants were healthy) the morphological variables would significantly change in response to altered life settings (i.e., home confinement). Indeed, apart from reduced physical activity, living conditions that could directly influence morphological status did not radically change [36]. Specifically, the availability of food and nutritional habits, which are major determinants of growth, have not changed dramatically during the pandemic. Moreover, some studies even noted a positive nutritional change in the studied period, with healthier food choices and regular meals during the pandemic, which was explained by increased attention regarding possible negative consequences of imposed measures [36]. Taken together, these results may explain our findings regarding the similarity in changes in anthropometric/morphological status in the non-pandemic and pandemic generations of adolescents.

### Limitations and Strengths

The most important limitation of this study is the small number of tests of muscular fitness included. However, as this is the preliminary analysis, we believe that this analysis will allow more detailed investigations in the future. Additionally, anthropometric/morphological status was evaluated using the most common measures (mass, height, body mass index), and further analyses are needed to take into account other measures of body composition.

This study is one of the first to directly investigate the influence of the COVID-19 lockdown on muscular fitness and anthropometric indices in adolescents. Although we are not able to provide the final conclusions regarding this problem, these findings provide a basis for future investigations. Additionally, the fact that all studied adolescents were selected from the same school (environment) and were evaluated by the same examiner are important strengths of the study. The results of this and similar studies should help to create an optimal program for training and exercise during the pandemic and similar situations where reduced movement opportunities occur.

## 5. Conclusions

This study demonstrated a decrease in muscular fitness of adolescents as a consequence of the COVID-19 lockdown, with a more significant decline among boys. Altogether, the decrease in fitness status may be connected to several reasons including a decrease of PAL during the pandemic, and reduced opportunities for systematic physical exercising (i.e., banned sport activities, online physical education classes).

Therefore, teachers, coaches, and other physical health-related professionals should develop adequate programs of exercise and training, especially for situations with movement restrictions. Considering the results of this study, special emphasis should be placed on the development of appropriate exercise programs for boys.

Further studies on the effects of the imposed lockdown measures on physical fitness are warranted. Special attention should be placed on information regarding the potential effects of the lockdown on body composition and aerobic endurance in children and youth.

## Figures and Tables

**Figure 1 healthcare-09-00681-f001:**
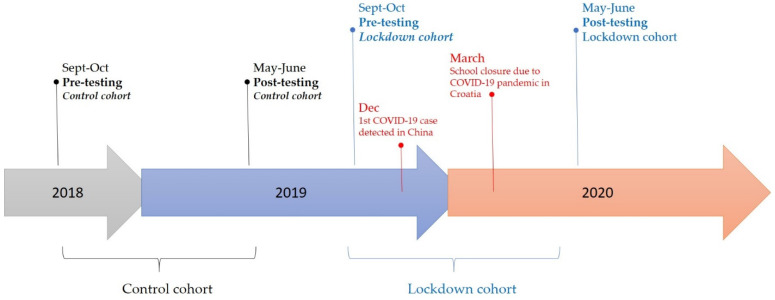
Timeline and design of the study.

**Table 1 healthcare-09-00681-t001:** Results of the repeated measurement factorial analysis of the variance with cohort (control vs. lockdown) as the group-factor, with F-test, *p*-level, and effect size (ES) reported.

Variables	Group (Lockdown vs. Control)			Measurement (Pre- vs. Post)			Interaction (Cohort x× Measurement)		
	F Test	*p*	ES	F Test	*p*	ES	F Test	*p*	ES
Body height	0.22	0.64	0.001	3.27	0.07	0.29	4.35	0.05	0.07
Body mass	0.12	0.73	0.001	4.37	0.04	0.04	0.59	0.44	0.001
Body mass index	0.23	0.63	0.002	0.16	0.69	0.001	0.91	0.34	0.001
Sit-ups	2.39	0.13	0.001	0.35	0.55	0.001	67.11	0.001	0.38
Run-600 m	0.08	0.77	0.001	40.68	0.001	0.28	61.97	0.001	0.37

**Table 2 healthcare-09-00681-t002:** Descriptive statistics (presented as means ± std. dev.) and post-hoc differences for study variables in the control and lockdown groups at baseline and follow-up.

Variables	Baseline		Follow-up	
	Control	Lockdown	Control	Lockdown
Body height (cm)	173.67 ± 9.58	173.14 ± 8.68	174.19 ± 9.66	173.27 ± 8.53
Body mass (kg)	64.69 ± 15.46	64.16 ± 12.15	65.99 ± 15.06	64.91 ± 12.16
Body mass index (kg/m^2^)	21.16 ± 3.73	22.44 ± 10.71	21.68 ± 3.81	21.53 ± 2.95
Sit-ups (rep)	36.52 ± 6.65	42.83 ± 11.33 ^¥^	44.04 ± 9.73 ^£^	33.89 ± 7.59 ^£, ¥^
Run-600 m (s)	173.18 ± 48.16	148.24 ± 21.38 ^¥^	168.78 ± 34.16	191.44 ± 36.38 ^£, ¥^

Note: ^£^ indicates significant (*p* < 0.05) post-hoc within group differences, ^¥^ indicates significant (*p* < 0.05) post-hoc between group differences, cm—centimeters, kg—kilograms, m—meters, rep—repetitions, s—seconds

**Table 3 healthcare-09-00681-t003:** Results of the repeated measurement factorial analysis of the variance (ANOVA) for the lockdown generation with gender (males vs. females) as the group-factor, with F-test, *p*-level, and effect size (ES) reported.

Variables	Group (Males vs. Females)			Measurement (Pre- vs. Post)			Interaction (Group × Measurement)		
	F Test	*p*	ES	F Test	*p*	ES	F Test	*p*	ES
Body height	83.02	0.001	0.57	0.05	0.83	0.001	0.001	0.96	0.001
Body mass	12.22	0.001	0.16	4.23	0.04	0.06	8.05	0.01	0.11
Body mass index	0.18	0.67	0.01	0.11	0.75	0.01	1.17	0.28	0.02
Sit-ups	16.05	0.001	0.21	45.44	0.001	0.43	11.08	0.01	0.15
Run-600 m	12.58	0.001	0.18	105.11	0.001	0.64	1.83	0.18	0.03

**Table 4 healthcare-09-00681-t004:** Descriptive statistics (presented as means ± std. dev.) and post-hoc differences for study variables for the lockdown generation.

Variables	Baseline		Follow-up	
	Males	Females	Males	Females
Body height (cm)	182.75 ± 5.38	168.87 ± 6.04	182.76 ± 4.94	168.84 ± 5.77
Body mass (kg)	70.10 ± 11.84	61.52 ± 11.46	73.19 ± 9.78	61.04 ± 11.46
Body mass index (kg/m^2^)	21.01 ± 3.31	23.08 ± 12.68	21.94 ± 2.64	21.34 ± 12.68
Sit-ups (rep)	51.25 ± 14.60	39.00 ± 6.76 ^¥^	35.33 ± 8.06 ^£^	33.22 ± 6.76 ^£^
Run-600 m (s)	129.17 ± 20.65	156.42 ± 15.91	179.41 ± 42.40	197.06 ± 15.91

Note: ^£^ indicates significant (*p* < 0.05) post-hoc within group differences, ^¥^ indicates significant (*p* < 0.05) post-hoc between group differences.

## Data Availability

Data are freely available here: http://www.kifst.unist.hr/~dado/index_files/mirela.sta (accessed on 5 June 2021).

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
