# Peer review of "Evidencing the Influence of the COVID-19 Pandemic and Imposed Lockdown Measures on Fitness Status in Adolescents: A Preliminary Report"

_healthcare, 2021, doi:10.3390/healthcare9060681_

Round 1
Reviewer 1 Report
Thank you for the opportunity to review this study – I enjoyed reading it. This study presents preliminary findings on changes in fitness among a small sample of Croatian adolescents across a school year. Two cohorts of students were examined – one of which was exposed to the social distancing measures enforced due to the COVID-19 pandemic. The study provides a very nice addition to the current literature base, due to its focus on changes in physical fitness, in addition to changes in physical activity. I thought the article was well-written, clear, and the conclusions were appropriate. I have some minor edits for the authors to consider:
Introduction
Line 47 – is it possible to add or clarify the units of measurement for the PAQ-A? I believe it is a score that takes into account engagement in different intensities of PA.
Line 76 – in the ‘aim’ statement, did the authors intend to say ‘investigate the impact of the COVID-19 lockdown on several DETERMINANTS of physical fitness’? I am not sure if the ‘determinants’ they refer to are the PA and anthropometric variables (if so, then determinants is fine). However, given that the dependent variables are the sit-up test and 600m run test, I wonder if the authors mean to refer to those variables – in which case, I think replacing the word ‘DETERMINANTS’ with the word ‘COMPONENTS’ would be appropriate.
Methods
Line 116 – Please could the research team review this sentence and tweak the language to improve clarity? I think the sentence is trying to convey that in Croatian schools, fitness is typically evaluated using a large range of physical tests. If this is the case perhaps the following is appropriate: In Croatian high-schools, physical fitness is typically evaluated using a wide range of physical fitness tests.
Lines 124-126 – I think this sentence could be clearer if the language were changed to the following: “Participants had to run a distance of 600 meters at a concrete athletic field, on a 300-meter oval track.”
Lines 128-129 – For clarity, please could the following language edits be incorporated: All tests were ADMINISTERED by AN experienced evaluator…
Line 140 – Please could the research team provide a reference to support the effect sizes that they are using.
Section 2.2 – do the research team have any data/references that indicate the reliability and/or validity of the fitness tests they administered? If so, it would be helpful to mention in this section.
Author Response
REVIEWER #1
Comments and Suggestions for Authors
Thank you for the opportunity to review this study – I enjoyed reading it. This study presents preliminary findings on changes in fitness among a small sample of Croatian adolescents across a school year. Two cohorts of students were examined – one of which was exposed to the social distancing measures enforced due to the COVID-19 pandemic. The study provides a very nice addition to the current literature base, due to its focus on changes in physical fitness, in addition to changes in physical activity. I thought the article was well-written, clear, and the conclusions were appropriate. I have some minor edits for the authors to consider:
RESPONSE: Thank you for your valuable comments and support. Also, thank you for recognizing the importance of our research. We tried to amend the manuscript according to your suggestions and hope that we adequately improved it.
Introduction
Line 47 – is it possible to add or clarify the units of measurement for the PAQ-A? I believe it is a score that takes into account engagement in different intensities of PA.
RESPONSE: Thank you for this suggestion. We added a short description of PAQ-A for clarifying scores we reported. Text now reads: “Specifically, the PAL of Croatian adolescents decreased from 2.97 ± 0.61 to 2.63 ± 0.68 (as measured by Physical Activity Questionnaire for Adolescents – PAQ-A) [14]. For clarity, PAQ-A is a questionnaire that assesses PAL during the last 7 days, with results ranging from 1 to 5, representing low and high PAL, respectively.”
Line 76 – in the ‘aim’ statement, did the authors intend to say ‘investigate the impact of the COVID-19 lockdown on several DETERMINANTS of physical fitness’? I am not sure if the ‘determinants’ they refer to are the PA and anthropometric variables (if so, then determinants is fine). However, given that the dependent variables are the sit-up test and 600m run test, I wonder if the authors mean to refer to those variables – in which case, I think replacing the word ‘DETERMINANTS’ with the word ‘COMPONENTS’ would be appropriate.
RESPONSE: Thank you for this observation. We indeed wanted to refer to sit-ups and run variables; thus, components seems more appropriate. We replaced word “determinants” with “components”. Text now reads: “Therefore, the aim of this study was to investigate the impact of the COVID-19 lock-down on several components of physical fitness in adolescents from Croatia.”
Methods
Line 116 – Please could the research team review this sentence and tweak the language to improve clarity? I think the sentence is trying to convey that in Croatian schools, fitness is typically evaluated using a large range of physical tests. If this is the case perhaps the following is appropriate: In Croatian high-schools, physical fitness is typically evaluated using a wide range of physical fitness tests.
RESPONSE: Thank you for this suggestion. We amended the text accordingly and now it reads: “In Croatian high-schools, physical fitness is typically evaluated using a wide range of physical fitness tests.”
Lines 124-126 – I think this sentence could be clearer if the language were changed to the following: “Participants had to run a distance of 600 meters at a concrete athletic field, on a 300-meter oval track.”
RESPONSE: Thank you for this suggestion. We changed the text accordingly and now it reads: “Participants had to run a distance of 600 meters at a concrete athletic field, on a 300-meter oval track.”
Lines 128-129 – For clarity, please could the following language edits be incorporated: All tests were ADMINISTERED by AN experienced evaluator…
RESPONSE: Thank you for this suggestion. Text now reads: “All tests were administered by an experienced evaluator, i.e., PE teacher – first author of the paper.”
Line 140 – Please could the research team provide a reference to support the effect sizes that they are using.
RESPONSE: Thank you for this suggestion. We added the following reference for effect sizes: Bakeman, R. Recommended effect size statistics for repeated measures designs. Behavior Research Methods 2005, 37, 379-384, doi:10.3758/bf03192707.
Section 2.2 – do the research team have any data/references that indicate the reliability and/or validity of the fitness tests they administered? If so, it would be helpful to mention in this section.
RESPONSE: Thank you for this suggestion. We added the references on metric characteristics of the tests we used. We added following references:
Tsigilis, N.; Douda, H.; Tokmakidis, S.P. Test-Retest Reliability of the Eurofit Test Battery Administered to University Students. Perceptual and Motor Skills 2002, 95, 1295-1300, doi:10.2466/pms.2002.95.3f.1295.
Neljak, B.; Novak, D.; Sporis, G.; Viskovic, S.; Markus, D. Metodologija vrednovanja kinantropoloskih obiljezja ucenika u tjelesnoj i zdravstvenoj kulturi [Methodology of evaluating the kinantropological characteristics of students in physical education: CROFIT norms]. Crofit norme. Zagreb, Kinezioloski fakultet 2012.
Reviewer 2 Report
The present study showed the changes in fitness status and compared between a control cohort and a Lockdown cohort. This issue is essential. My major concerns are as follows.
- Participants and study design subsection (2.1) showed that the present study was part of another longitudinal investigation. What was the purpose of the original research and the inclusion criteria of the participants? Show the additional information of the original study.
- Show the detail of the COVID-19 pandemic in Croatia. How long did the lockdown in March 2020 last? During the lockdown, could people go out for shopping or exercise?
- In results, mean changes and 95%CIs are preferable to describe in text. The present manuscript was overlapping in the text and the table.
- In tables, the authors should select more suitable symbols.
- In the Conclusions, the authors should delete the description about PAL. The present study did not assess the PAL in this cohort. Also, the current conclusion was redundant. A more concise conclusion is desirable.
Author Response
REVIEWER #2
Comments and Suggestions for Authors
The present study showed the changes in fitness status and compared between a control cohort and a Lockdown cohort. This issue is essential. My major concerns are as follows.
RESPONSE: Thank you for recognizing the importance of our research and providing the opportunity to improve the manuscript. We tried to amend the manuscript according to your valuable comments and suggestions.
- Participants and study design subsection (2.1) showed that the present study was part of another longitudinal investigation. What was the purpose of the original research and the inclusion criteria of the participants? Show the additional information of the original study.
RESPONSE: Thank you for this insightful suggestion. Brief description of the original study is now added, and the text reads: “This study was part of another longitudinal investigation which initiated in 2017, and all participants were informed about the aims and protocols of the study, and their parents signed informed consent forms. The original study titled “Physical activity, substance misuse, and factors of influence in adolescence” aimed to establish changes in health-related variables (i.e., PAL, misuse of psychoactive substances, fitness status), which allowed us to observe adolescents even during the COVID-19 pandemic.”
- Show the detail of the COVID-19 pandemic in Croatia. How long did the lockdown in March 2020 last? During the lockdown, could people go out for shopping or exercise?
RESPONSE: Thank you for this suggestion, it will certainly improve our manuscript. Text now reads: “The strictest lockdown in Croatia was implemented from 19th of March and lasted till 27th of April 2020. Schools and universities were closed, public gatherings were prohibited, cafes, restaurants, fitness centres, shopping centres and churches were closed. Playground and parks were also closed, but there was no strict prohibition of outdoor individual training (i.e., running, cycling). However, schools were closed till the start of a new school year, in September 2021.”
- In results, mean changes and 95%CIs are preferable to describe in text. The present manuscript was overlapping in the text and the table.
RESPONSE: Thank you for this valuable suggestion. We reported the mean changes and 95%CI in the text of the Results section. Text now reads:
“Descriptive statistics evidenced that lockdown group achieved lower results in fitness tests (e.g. Sit-ups and Run-600m) compared to control group at follow-up testing. Post-hoc analysis indicated differential changes in sit-ups and run 600m between cohorts. In brief, both variables positively changed in Control group from baseline to follow-up testing (mean change; 95%CI: Sit-ups: 7.79; 6.05-9.53; Run-600m: 4.40; -4.26-13.06). On the other hand, the negative changes were evidenced for Lockdown group in Sit-ups (mean change; 95%CI: 7.59; 5.51-9.67) and Run-600m (mean change; 95%CI: 41.97; 33.99-49.95) (Table 2).”
and
“Post-hoc analyses indicated significant decrease from baseline to follow-up testing of sit-ups in females (mean change; 95%CI: 5.59; 3.27-7.91), with more evident decrease in males (mean change; 95%CI: 12.0; 8.13-15.87) in Lockdown generation (Table 4).”
- In tables, the authors should select more suitable symbols.
RESPONSE: Thank you for this suggestion. We changed symbols to superscript format.
- In the Conclusions, the authors should delete the description about PAL. The present study did not assess the PAL in this cohort. Also, the current conclusion was redundant. A more concise conclusion is desirable.
RESPONSE: Thank you for this suggestion. We amended the Conclusion and now it reads: “This study demonstrated a decrease in muscular fitness of adolescents as a con-sequence of the COVID-19 lockdown, with a more significant decline among boys. Al-together, the decrease in fitness status may be connected to several reasons including decrease of PAL during pandemic, and reduced opportunities for systematic physical exercising (i.e. banned sport activities, “online” physical education classed).
Therefore, teachers, coaches, and other physical health-related professions should develop adequate programs of exercise and training, especially for situations with movement restrictions. Considering the results of this study, special emphasis should be placed on the development of appropriate exercise programs for boys.
Further studies on effects of the imposed lockdown measures on physical fitness are warranted. Special attention should be placed on information regarding potential effects of lockdown on body composition and aerobic endurance in children and youth.”